# In-silico analysis of myeloid cells across the animal kingdom reveals neutrophil evolution by colony-stimulating factors

Damilola Pinheiro*, Marie-Anne Mawhin, Maria Prendecki, Kevin J Woollard*

Centre for Inflammatory Disease, Department of Immunology and Inflammation, Imperial College London, London, United Kingdom

**Abstract** Neutrophils constitute the largest population of phagocytic granulocytes in the blood of mammals. The development and function of neutrophils and monocytes is primarily governed by the granulocyte colony-stimulating factor receptor family (CSF3R/CSF3) and macrophage colony-stimulating factor receptor family (CSF1R/IL34/CSF1) respectively. Using various techniques this study considered how the emergence of receptor:ligand pairings shaped the distribution of blood myeloid cell populations. Comparative gene analysis supported the ancestral pairings of CSF1R/IL34 and CSF3R/CSF3, and the emergence of CSF1 later in lineages after the advent of Jawed/Jawless fish. Further analysis suggested that the emergence of CSF3 lead to reorganisation of granulocyte distribution between amphibian and early reptiles. However, the advent of endothermy likely contributed to the dominance of the neutrophil/heterophil in modern-day mammals and birds. In summary, we show that the emergence of CSF3R/CSF3 was a key factor in the subsequent evolution of the modern-day mammalian neutrophil.

## Introduction

Phagocytes are key effector immune cells responsible for various biological processes; from orchestrating responses against invading pathogens to maintaining tissue homeostasis and neutrophils are the most abundant population of granulocytic phagocytes present in mammalian blood (*Adrover et al., 2019*; *Hidalgo et al., 2019*; *Ng et al., 2019*; *Yvan-Charvet and Ng, 2019*; *Evrard et al., 2018*). Neutrophil and heterophils (a functionally analogous granulocytic phagocyte population present in non-mammals and some mammals; *Montali, 1988*) arise from a shared pool of haematopoietic stem cells and mitotic myeloid progenitor cells that can also differentiate into monocytes, eosinophils and basophils following exposure to the relevant growth factor (*Adrover et al., 2019*; *Yvan-Charvet and Ng, 2019*; *Mehta et al., 2014*). The development and life cycle of mammalian neutrophils through a continuum of multipotent progenitors to a post-mitotic mature cell has been well described and has been recently reviewed (*Hidalgo et al., 2019*; *Yvan-Charvet and Ng, 2019*).

The development and function of myeloid phagocytes is mediated through lineage-specific transcription factors and pleiotropic glycoproteins - termed colony-stimulating factors (CSFs)- acting in concert on myeloid progenitor cells. CSF1, CSF3, and their cognate receptors, are lineage-specific and responsible for the differentiation and function of monocytes/macrophages and neutrophils, respectively (for full nomenclature see *Supplementary file 1*; *Dai et al., 2002*; *Yoshida et al., 1990*; *Umeda et al., 1996*; *Liu et al., 1996*; *Lieschke et al., 1994*). There is a large body of evidence demonstrating the requirement of CSFs for cell development as multiple studies in knockout mice have shown that CSF1R/CSF1 and CSF3R/CSF3 are linked to the development of monocytes and neutrophils in vivo. The loss of CSF3 or CSF3R directly affects neutrophil populations resulting in a severe neutropenia, but not the complete loss of mature neutrophils in the models studied (*Liu et al.,*

**\*For correspondence:**
d.pinheiro@imperial.ac.uk (DP);
k.woollard@imperial.ac.uk (KJW)

**Competing interests:** The authors declare that no competing interests exist.

*1996*; *Lieschke et al., 1994*; *Basu et al., 2000*). The loss of CSF1 caused reduced cavity development of the bone marrow, loss of some progenitor populations, monocytopenia, and a reduced population of neutrophils in the bone marrow, although interestingly, elevated levels of neutrophils were observed in the periphery (*Dai et al., 2002*; *Yoshida et al., 1990*; *Hibbs et al., 2007*).

Similarly, in humans, single gene mutations have been described in both CSF3 and CSF3R resulting in severe congenital neutropenia (SCN). Furthermore, mutations in ELANE, a gene that encodes for the neutrophil granule serine protease -neutrophil elastase- and acts as a negative regulator of CSF3R signalling can also result in SCN (*Piper et al., 2010*; *Hunter et al., 2003*; *Garg, 2020*; *Nayak et al., 2015*). In contrast to some animal models, individuals present as children with early onset life-threatening infections because they lack mature neutrophils in the circulation as the neutrophil progenitors in the bone marrow do not progress beyond the myelocyte/promyelocyte stage (*Lakshman and Finn, 2001*). These studies demonstrate that CSF receptor/ligand pairings are essential for homeostatic neutrophil development and are intrinsically linked with neutrophil function, arguably making them ideal surrogates to study neutrophil evolution. Through multiple methods we examined the emergence of the respective CSF ligand and receptor genes and proteins across the Chordate phylum and demonstrated how CSF1R/CSF1 and CSF3R/CSF3 pairings contributed to the evolutionary adaptations of the mammalian neutrophil.

## Results

### The neutrophilic/heterophilic granulocyte is the predominant granulocyte in the blood of mammals and aves

The presence of analogous myeloid granulocytes and agranulocytes was examined by comparing available complete blood count (CBC) data, where applicable, for various animal orders and demonstrated the possible distribution of myeloid cells in blood across evolution. The earliest chordates were represented by two groups, the first; jawed fish ($4.10 \times 10^8$ million years ago [MYA]), which included; coelacanths, elasmobranchs (Whale shark) and chimaeras (Australian Ghost sharks). Jawless fish containing lampreys and hagfish ($3.6 \times 10^8$ MYA) represented the second group. The next group represented chronologically was the amphibians ($3.5 \times 10^8$ MYA); containing anurans and gymnophiona. The reptilian class was represented by three different orders, squamata ($3.3 \times 10^8$ MYA), crocodilia ($2.4 \times 10^8$ MYA) and testudines ($2.1 \times 10^8$ MYA). The following group was the closely related avian class ($7.0 \times 10^7$ MYA), which, contained birds from both paleognaths and neognaths. The final class of chordates evaluated was Mammalia, where all three orders; monotremata ($1.1 \times 10^8$ MYA), marsupial ($6.5 \times 10^7$ MYA) and placental ($6.25 \times 10^7$ MYA) were represented. All the species examined for cell distribution and for the subsequent gene sequence and/or protein homology studies were visualised in a species tree (*Figure 1a*).

All classes within the phylum Chordata were evaluated for populations of neutrophils/heterophils; eosinophils and basophils. Azurophils, a specialised population of granulocytes – analogous to both the mammalian neutrophil and monocyte- but unique to reptiles were also included. A meta-analysis was performed on representative aggregated data to show absolute counts per cell type per order and finally, the proportional composition of myeloid cells only per cell type (*Figure 1b*).

The presence of all myeloid cells was confirmed in at least one representative species from each order examined. Jawed fish, which comprised sharks and rays, were unique in having both heterophils ($2.0 \times 10^9$/L) and neutrophils ($0.045 \times 10^9$/L) in the shark. All other orders had either neutrophilic or heterophilic populations only. Heterophils formed a sizeable population of granulocyte cells in Amphibians (the order of Anura; $2.9 \times 10^9$/L), Reptiles (the orders; squamata, $6.9 \times 10^9$/L; testudines, $2.1 \times 10^9$/L; and crocodiles, $3.2 \times 10^9$/L) and Aves ($12.7 \times 10^9$/L). In contrast, neutrophils were the majority population in all three mammalian orders (monotremata, $6.0 \times 10^9$/L; marsupalia, $5.3 \times 10^9$/L; and placental, $5.2 \times 10^9$/L [*Figure 1bi*]). Eosinophils and basophils were over-represented in the orders of amphibians, reptiles and aves, when compared to mammals, where they constituted minority populations [*Figure 1bi*].

Analysis of the proportional composition of total myeloid white blood cells (WBC) showed interesting changes in the distribution of granulocytes across the different species classes. Jawed fish lineages, represented by sharks and rays, retained both a majority heterophil population (49.0%) and a minority neutrophil population (1.6%). The cold-blooded orders; anura (29.7%), squamata (55.3%),

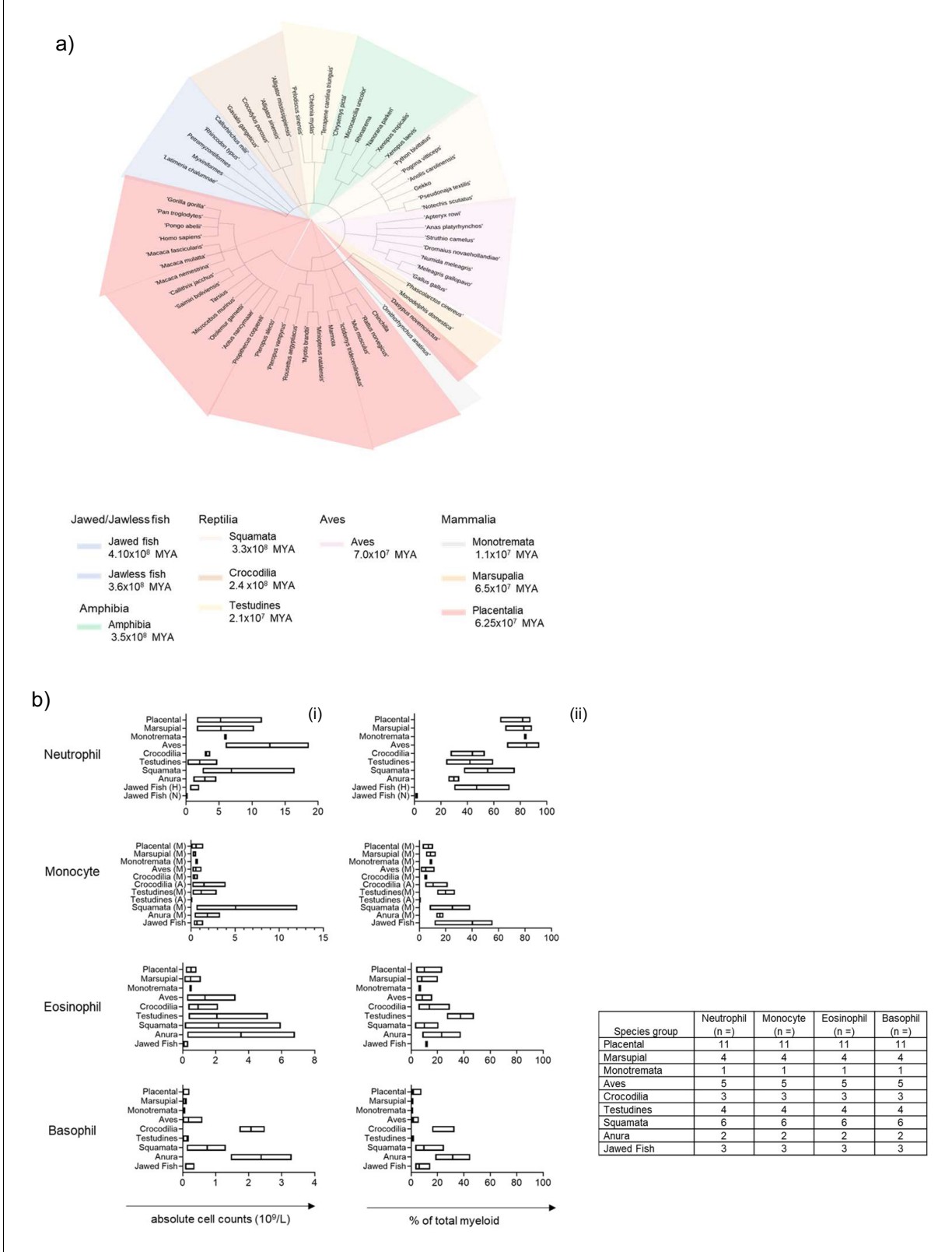

**Figure 1.** Population comparison of blood myeloid cell subset distribution in chordates demonstrates predominance of the neutrophilic granulocyte in birds and mammals. (A) Species tree of the animals within Phylum Chordata examined, sub-classified by animal order or class. (B) Meta-analysis of aggregated phylum Chordata complete blood cell counts (excluding all lymphoid cells). Data is visualised as a floating bar and the line represents the

*Figure 1 continued on next page*

*Figure 1 continued*

mean value and shows absolute counts per cell type per order (i) and composition of myeloid cells per cell type (ii). (N) = Neutrophils, (H) = Heterophils, (A) = Azurophils, (M) = Monocytes.

testudines (41.9%) and crocodilia (43.7%); retained a large population of neutrophils/heterophils; although proportions of the population varied within related animal orders and between the different species classes (*Figure 1bii*). In contrast, within the warm-blooded classes and orders; aves (84.7%), monotremata (83.8%), marsupial (82.6%) and placental (81.8%) the opposite distribution was observed and the neutrophil/heterophil was the most abundant peripheral blood cell type (*Figure 1bii*). Interestingly, basophils and eosinophils were well represented in the anuran order (31.5%, and 23.2%, respectively). A similar pattern was observed in the reptilian lineage; however, there were differences between orders as squamata (10.1%) and crocodilia (13.8%) had lower proportions of eosinophils compared to testudines (37.5%). Conversely, basophils were elevated in crocodilia (27.2%) compared to squamata and testudines (9.7% and 0.6% respectively) [*Figure 1bii*]. Suggesting that within closely related orders that share similar habitats or environments, individual species favoured different configurations of myeloid WBCs. Eosinophils were present as a small population in the mammalian lineages, again variation between the respective classes was observed as monotremata had a smaller proportion of cells (6.7%) compared to the marsupial and placental orders, which had similar values (8.2% and 10.0% respectively). In contrast, compared to all the other classes, basophils seemed to have been largely lost from the mammalian classes of monotremata (0.7%) and marsupials (0.6%), with a minority population observed in placental animals (1.7%) [*Figure 1bii*].

Monocytes are blood agranulocytic cells that are closely related to neutrophils sharing both growth and survival factors, early development pathways with common progenitors, and in some instances, effector functions in the respective terminally differentiated cells (*Evrard et al., 2018*; *Yáñez et al., 2017*; *Halene et al., 2010*). Interestingly, the monocyte population peaked both in terms of number and proportion within the reptilian class, in the orders; squamata ($5.06 \times 10^9$/L and 25% respectively) and testudines ($1.2 \times 10^9$/L and 19.8% respectively). The azurophil – a specialised myeloid cell population- peaked in the crocodilian lineage ($1.5 \times 10^9$/L and 10.5% respectively), however; it remains unclear whether to classify the azurophil as a distinct cell type, as we have done, or as a subset of monocyte or neutrophil. Aves is the most closely related phylogenetic group to reptiles as they share a recent common ancestor. Although aves has lost the azurophil, they retain a small population of monocytes (4.9%). Interestingly, a similar proportion is observed in the more distantly-related mammalian class, across all three orders (monotremata (8.8%), marsupial (8.5%) and placental (6.9%)), suggesting that the advent of endothermy may have played a role in the distribution of monocytes in warm-blooded animals [*Figure 1bii*] In summary, we show in phylum Chordata that the majority of peripheral myeloid blood cells are comprised of granulocytes, although the proportion of basophils, eosinophils and neutrophils varies according to the respective orders and lineages. By the advent of mammals and birds however, the neutrophil has become the predominant granulocyte of the blood.

## Interrogation of CSF1/CSF1R and CSF3/CSF3R and C/EBP gene family reveals their loss in the early lineages

The development and maturation of a neutrophil from a multipotent progenitor to a lineage committed post-mitotic cell is driven by CSF3 working in concert with a number of transcription factors, including Pu.1, GFI1 and Runx, whose roles have been well established (*Yvan-Charvet and Ng, 2019*; *Halene et al., 2010*). One of the most heavily involved family of transcription factors in neutrophil development is the CCAT enhancer binding-protein family (C/EBP), comprising six members (C/EBPα, β, δ, γ, ε, and τ) (*Halene et al., 2010*; *Zhang et al., 1997*; *Yamanaka et al., 1997*; *Lekstrom-Himes and Xanthopoulos, 1999*). Multiple studies have shown that C/EBPα, C/EBPβ, and C/EBPε are required for different stagesof granulopoiesis and are further detailed in *Supplementary file 2*. Briefly, under steady state conditions *c/ebpα*$^{-/-}$ mice are neutropenic as there is a block in the maturation stage between the common myeloid progenitor (CMP) and the granulocyte- monocyte progenitor (GMP) (*Smith et al., 1996*; *Zhang et al., 2004*). In contrast, *c/ebpε*$^{-/-}$

mice lack functionally mature neutrophils as the stage between GMP and terminal differentiation is blocked (*Morosetti et al., 1997*; *Chih et al., 1997*). Thus, demonstrating that *c/ebpα* and *c/ebpε* have distinct roles in the early (myeloid progenitor progression) and later (maturation) stages of neutrophil development (*Halene et al., 2010*; *Zhang et al., 1997*; *Yamanaka et al., 1997*; *Lekstrom-Himes and Xanthopoulos, 1999*; *Smith et al., 1996*; *Zhang et al., 2004*; *Morosetti et al., 1997*; *Chih et al., 1997*). *C/ebpβ* is required during emergency granulopoiesis as deficient mice could not mobilise a response to systemic fungal infection or cytokine stimulation, even though steady state granulopoiesis remained unaltered (*Zhang et al., 2002*; *Hirai et al., 2006*; *Screpanti et al., 1995*; *Tanaka et al., 1995*). Having established, the presence of the neutrophil/heterophils across the phylum Chordata, we then evaluated the presence of CSFR/CSF and C/EBP genes in the different animal classes to further understand the emergence of granulopoiesis in phylum Chordata.

Gene data for fifty-nine species was collected from the NCBI Gene and Ensembl databases for the following genes; CSF1R, IL34, CSF1, CSF3R, CSF3, C/EBPα, C/EBPβ and C/EBPε and used to generate a heat map. Interestingly, CSF1, CSF1R, CSF3R and CSF3 are absent from the cartilaginous fishes; the Australian Ghost shark and the Whale shark (*Figure 2a*). In contrast, the Coelacanth - another early lineage - retained CSF1R, IL34, CSF3R and CSF3 genes. CSF1 was absent from all the jawed fishes although IL34 was present. CSF1R was the only the receptor present in both the

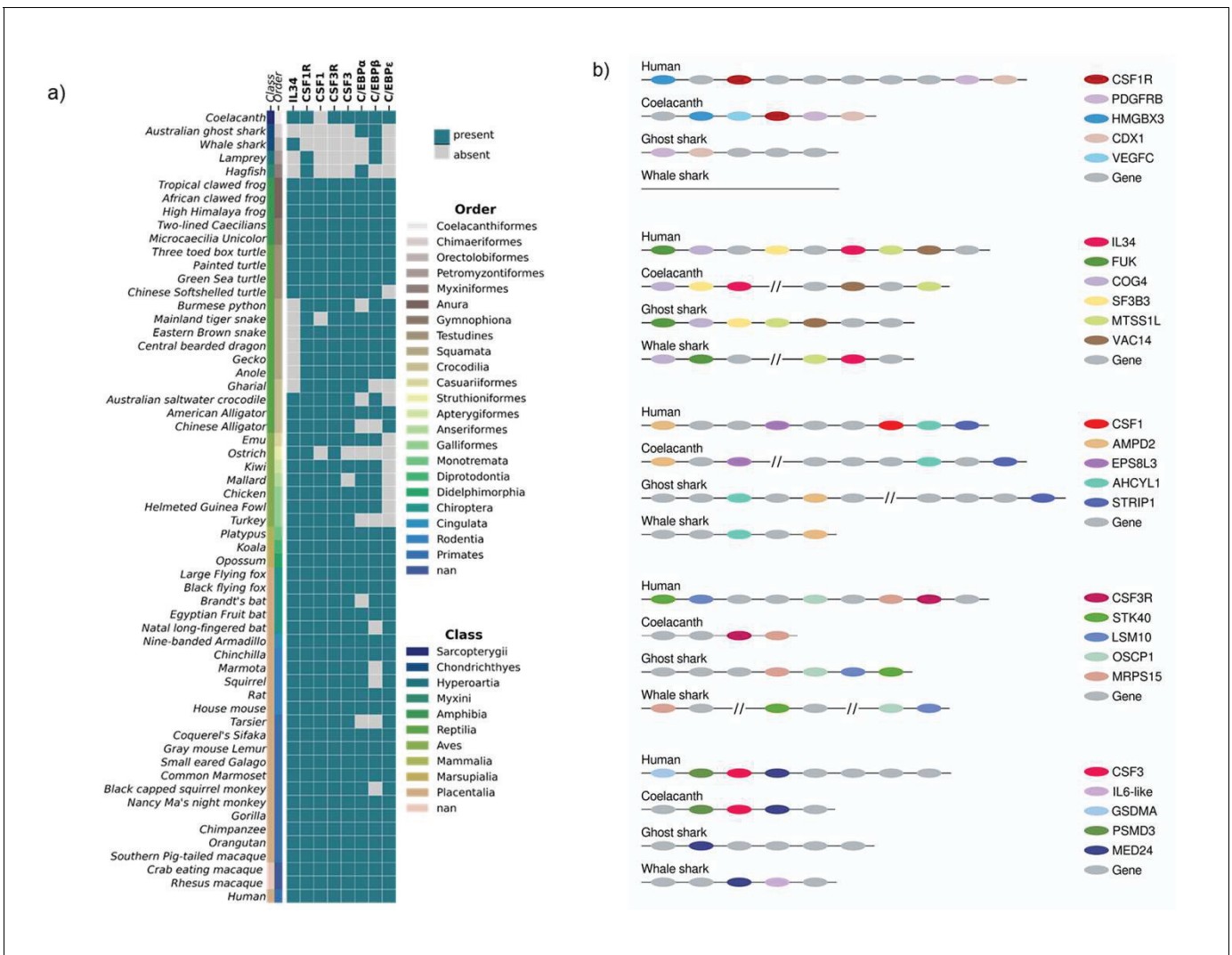

**Figure 2.** Analysis of chordate CSF1/CSF1R, CSF3/CSF3R, and neutrophil-related transcription factors reveals their loss in Jawed/Jawless fish lineages. (A) Heat map of CSF1R/IL34/CSF1, CSF3R/CSF3, and neutrophil-related transcription factors in selected members of Phylum Chordata. (B) Syntenic maps of CSF1R/IL34/CSF1 and CSF3R/CSF3 in selected Jawed fish compared to human.

lamprey and hagfish, and CSF3R and all other ligands were absent (*Figure 2a*). The Jawed/Jawless fish accounted for the largest loss of genes within in an animal order as generally; CSF1R, IL34, CSF1, CSF3R, and CSF3 were largely all present in amphibia, reptilia, aves, and mammals. The exception being in the reptilian order of squamata, where IL34 had been lost. There were also examples of gene loss at the species level, CSF1 and CSF3 were absent from the Ostrich; however, they had not been lost overall in from the group (*Figure 2a*).

During granulocytic differentiation, there is temporal expression of C/EBP gene family members according to the maturation stage of the neutrophil. C/EBPα is expressed highest in early progenitors and decreases through differentiation to low levels of expression in mature granulocytes. The inverse expression pattern is seen with CEBPε, where it is highly expressed in mature neutrophils and there's minimal expression in progenitors. C/EBPβ follows a different pattern with expression increasing in progenitor populations and levels being maintained through to maturation (*Scott et al., 1992*). Given the importance of the three members to mammalian neutrophil development we considered the conservation of these members across phylum Chordata. Interestingly, within the early Jawed/Jawless lineages there had been local loss of C/EBPα, C/EBPε or both from some members, however all fish examined retained C/EBPβ and the Coelacanth had all three genes, indicative of their presence in a shared common ancestor (*Figure 2a*). The analysis showed very strongly supported evolutionary conservation of the C/EBP gene family members in tetrapod lineages, as almost every group had a minimal combination of C/EBPα or C/EBPβ and C/EBPε (the exception being the aves class where CEBP/ε was absent). Interestingly, within the aves class both ostriches and turkeys lacked all the gene family members examined. Taken together, these results suggested that there were local factors determining the retention and loss of genes as well as a strong level of redundancy among the C/EBP gene family. A high degree of functional redundancy between family members has been well described and is supported by various knockout studies in mice. These show that at different stages of granulopoiesis, C/EBP gene family members - particularly C/EBPβ- can in part compensate functionally for each other (*Jones et al., 2002*; *Akagi et al., 2010*).

Interestingly, within the early lineages of the jawed fishes, IL34 was observed in two species and the Coelacanth was the only species to have CSF1R, CSF3R and CSF3 present. As these three distinct lineages share a common ancestor, this would indicate that there had been local gene loss in both the Whale shark and Australian Ghost shark. CSF1 was absent from all lineages and it was unclear as to whether this absence was due to loss or that CSF1 had not yet evolved. To further address this, we used a syntenic approach (*Engström et al., 2007*), that takes advantage of a process where genes that have co-evolved together physically co-localise within the loci, and using this to manually map out the orthologous gene locations in the Jawed fish. Syntenic maps were generated for the CSF1R (CSF1R, IL34, CSF1) and CSF3R (CSF3R and CSF3) family of genes using the human orthologue as a reference point (*Figure 2b*).

In humans, the CSF1R gene is located downstream of the HMGBX3 gene and upstream of PGDFRB (a gene paralogue of CSF1R) and CDX1. A similar orientation is observed in the Coelacanth, where CSF1R is downstream of HMGBX3 and adjacent to PDGFRB and CDX1. The orientation is inversed in the Australian Ghost shark, where PDGFRB and CDX1 co-localise together upstream but HMGBX3 and CSF1R have been lost (*Figure 2b*). Interestingly in the Whale shark, CSF1R and all the flanking genes are absent, suggesting that this section in its entirety may have been lost (*Figure 2b*). Human IL34 co-localises with FUK, COG4, and SF3B3 upstream, and the MTSS1L and VAC14 genes immediately downstream. Both the Coelacanth and the Whale shark have partially retained the syntenic combination but not in the same location. IL34 is situated immediately downstream of COG4 and SF3B3, whereas MTSS1L and VAC14 are located elsewhere on the Coelacanth chromosome (*Figure 2b*). In the Whale shark, IL34 is immediately adjacent to MTSS1L and these genes are both downstream of FUK and COG4, which have co-localised next to each other (*Figure 2b*). The gene arrangement of the Australian ghost shark most closely resembles the human, as FUK, COG4, SF3B3, MTSS1L and VAC14 all co-localise to the same region on the chromosome and the absence of IL34, suggest it has been loss in the process of a local gene rearrangement (*Figure 2b*). The CSF1 human orthologue has formed a contiguous block with AHCYL1 and STRIP1 and is flanked upstream by AMPD2 and EPS8L3. Interestingly although CSF1 is absent from all members, the flanking genes have largely been retained. Similar to the human arrangement, in the Coelacanth; AMPD2 and EPS8L3 co-localised together, while AHCYL1 and STRIP1 are located in close

proximity in a different location (*Figure 2b*). In both the Australian Ghost shark and Whale Sharks, AHCYL1, and AMPD2 are located together, STRIP1 is located elsewhere in the Australian Ghost shark and has been lost entirely from the Whale shark (*Figure 2b*). These lineages arose early in evolution but have very similar synteny structure to the later emerging human chromosome, suggesting that the CSF1 gene entered this location in an ancestor that emerged after the jawed fish.

A similar pattern emerged in the CSF3 family of proteins when analysed. In humans, the CSF3R gene is located downstream of STK40, LSM10, OSCP1, and MRPS15. The Coelacanth is the only species to retain CSF3R and that is located immediately adjacent to MRPS15, however STK40, LSM10, and OSCP1 have been lost. In contrast, although both the Whale shark and Australian Ghost shark have presumably lost the CSF3R gene, they have retained the four other flanking genes, either in one location as in the Australian Ghost shark, or distributed along the chromosome, as in the Whale Shark (*Figure 2b*). Human CSF3 co-localises downstream of GSDMA and PSMD3 and is adjacent to MED24. Again, a similar arrangement is observed in the Coelacanth, with CSF3 flanked by PSMD3 and MED24 upstream and downstream respectively. As before, the Whale shark and Australian Ghost shark are similar in their gene arrangements as GSDMA and PSMD3 have been lost, and the only gene retained in the Australian Ghost shark is MED24 (*Figure 2b*). Intriguingly, in the Whale shark, MED24 co-localises with an IL6-like gene, which could be a functional paralogue of CSF3 (*Figure 2b*). These results suggest there is more than one receptor-ligand family involved in the development and maturation of heterophils and neutrophils in the early lineages. Taken together, these data suggest that CSF3R/CSF3 and CSF1R were present in a common ancestor to early lineages and there have been some local losses in selected members. CSF1 appears to have evolved independently of its cognate receptor, after the emergence of Jawed lineages and prior to the advent of the tetrapod lineages.

## Analysis of chordate orthologous protein homology further supports the ancestral pairing of CSF1R/IL34 and CSF3R/CSF3 in early lineages

The syntenic analysis established the presence of the CSF1R and CSF3R gene families in Chordates. Interestingly, CSF1R/IL34 and CSF3R/CSF3 had already evolved by the emergence of Chordates, as evidenced by their existence in some of the early lineages of Jawed/Jawless fish. However, the gene data in isolation did not provide a complete understanding and additional analysis was needed. Orthologous proteins are considered to have the same function in different species and therefore, it's broadly assumed that the proteins will largely be conserved at the primary and structural levels (*Forslund et al., 2011*). To further elucidate the evolutionary process, we compared the shared sequence similarity of the orthologous CSF1R and CSF3R protein families, as this data is widely available. The protein sequences for the following proteins; CSF1R, IL34, CSF1, CSF3R, and CSF3 for fifty-nine species were collated from the NCBI and Ensembl databases. The shared sequence similarity for each individual sequence was generated using the NCBI BLAST engine tool and the relevant human orthologue submitted as the query. The resulting data was then plotted in groups as identified by animal order and visualised in a bar chart (*Figure 3a*).

The level of shared sequence similarity observed in the receptors, and therefore conservation of the primary protein sequence, varied across the different animal orders. The highest levels of shared sequence conservation for both CSF1R and CSF3R were observed in the placental mammals (CSF1R; 86.7%, CSF3R; 81.7), which would be anticipated as the human is a member of this group. The lowest level of shared sequence similarity was observed in the Jawed/Jawless fish (CSF1R; 43.1%, CSF3R; 33.0%) (*Figure 3a*). Interestingly, the CSF1R and CSF3R protein sequence values from early mammals, monotremata (CSF1R; 60.4%, CSF3R; 46.9%) and marsupalia (CSF1R; 60.4%, CSF3R; 51.0%), ranged between the earlier orders; aves (CSF1R; 51.5%, CSF3R; 39.4%), testudines (CSF1R; 53.4%, CSF3R; 41.2%), crocodilia (CSF1R; 52.6%, CSF3R; 37.6%), squamata (CSF1R; 47.1%, CSF3R; 37.2%), amphibia (CSF1R; 47.2%, CSF3R; 35.7%) and placental mammals (*Figure 3a*). While there is not a clear consensus as to what the minimum percentage of shared sequence similarity needed to correlate with conservation of function is, it is notable that in this dataset for both CSF1R and CSF3R - independently of each other - the baseline value of shared sequence similarity was approximately 40% (*Figure 3a*).

The syntenic analysis also demonstrated that of the three ligands, IL34 emerged the earliest and this was reflected in the protein data. As observed with the receptors, the highest level of shared sequence similarity is in the placental mammal group (83.3%), and the lowest in the jawed/jawless

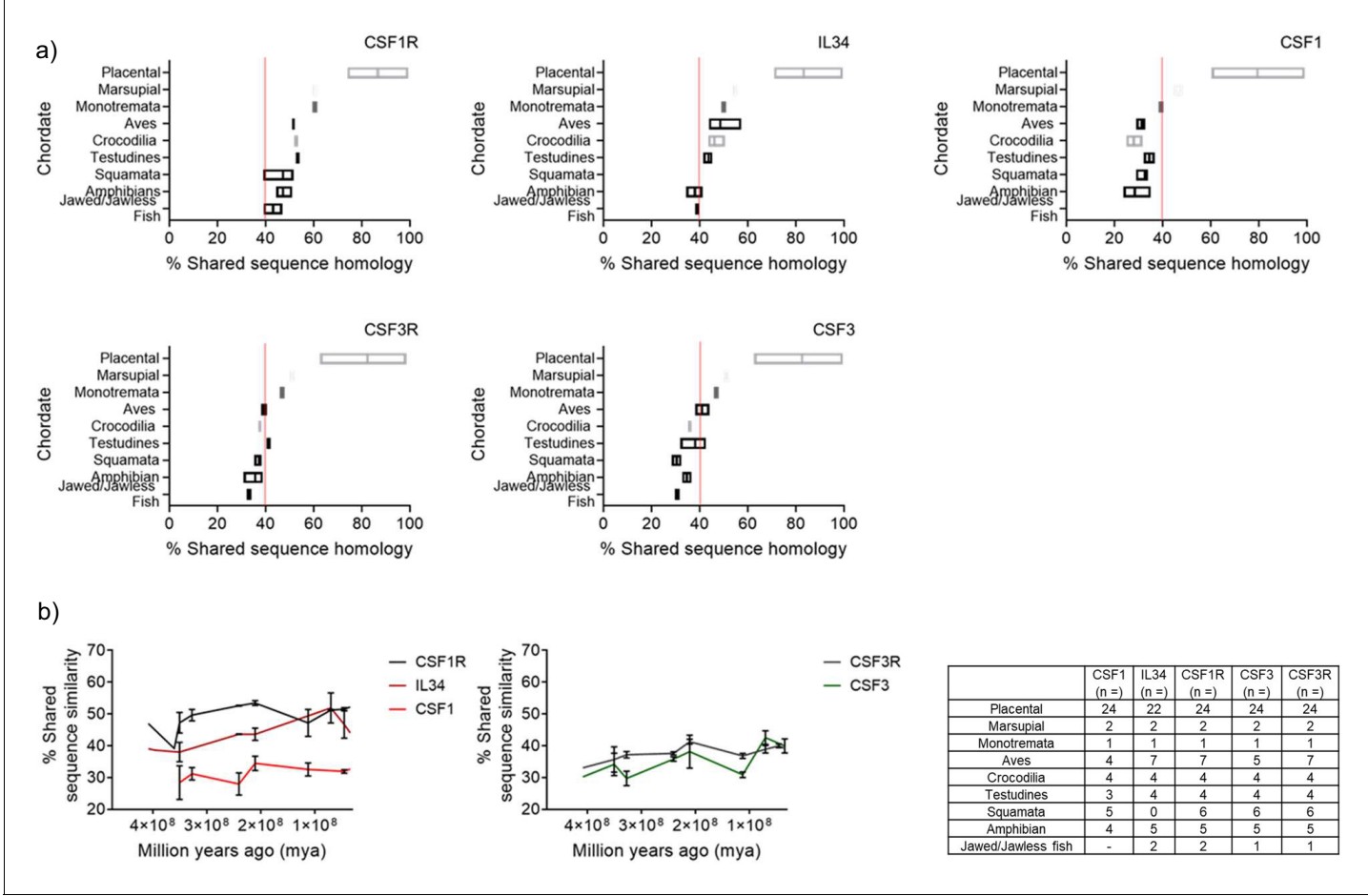

**Figure 3.** Shared sequence similarity analysis of Chordate CSFR/CSF protein homology further supports the ancestral pairing of CSF1R/IL34 and CSF3R/CSF3 in early lineages. (A) Representative plots of % shared sequence similarity for CSF1R, CSF3R, IL34, CSF1, and CSF3 in the respective sub-groups of Phylum Chordata, data is visualised as a floating bar and the line represents the mean value. Baseline shared similarity is represented by the red line. (B) Graphical representation of % shared sequence similarity in sub-groups of Phylum Chordata *versus* time.

fish (38.8%) (*Figure 3a*). The IL34 protein sequence values from early mammals; monotremata (50.0%) and marsupalia (54.8%), grouped very closely with the earlier orders of; aves (48.5%), crocodilia (46.1%), testudines (43.6%), amphibia (38.0%) and the jawed/jawless fishes (*Figure 3a*). IL34 was absent from the members of squamata examined. CSF3 is the next best conserved ligand, which again agreed with the synteny data. The highest level of shared sequence similarity was in the placental mammal group (84.2%), and the lowest in the jawed/jawless fish (30.7%) (*Figure 3a*). The CSF3 protein sequence values were spread among the orders from monotremata (57.2%) and marsupalia (61.1%) to aves (41.0%), crocodilia (35.4%), testudines (39.7%), squamata (30.4%) and amphibia (32.9.1%) (*Figure 3a*). Finally, CSF1, which is completely absent in jawed/jawless fish, had similar values to CSF3, in terms of shared sequence similarity across the respective groups of; placental mammals (79.4%), monotremata (39.4%), marsupalia (46.5), aves (31.3%), crocodilia (28.0%), testudines (34.5%), squamata (32.6%) and amphibia (28.4%). Intriguingly, the 40% baseline is applicable to the ligand data. In IL34, which is the oldest ligand, the shared sequence similarity for the majority of the groups was above 40%. Whereas for both CSF3 and CSF1, which emerged later, in the majority of the groups the shared sequence similarity was under 40% (*Figure 3a*).

The analysis of orthologous CSF1R and CSF3R protein families illustrated that the mammalian proteins- largely within the placental mammals- had changed considerably compared to the other lineages and therefore were excluded from subsequent analysis. To further interrogate how the respective CSF1R and CSF3R families co-evolved in the early lineages, the shared sequence similarities for each order were plotted against time (*Figure 3b*). Interestingly, the trajectories for CSF1R

and IL34 and CSF3R and CSF3, largely tracked to each other. This suggested that they evolved at a similar pace across the same period of time and is indicative of evolutionary pressure restricting changes to cognate receptors and ligands. As expected, CSF1 did not have the same restrictive pattern in the earlier lineages as it emerged later (*Figure 3b*). These results supported the early emergence of CSF3R, CSF1R, and IL34. Interestingly, although CSF3 developed later than CSF3R, the two have co-evolved in step together. In contrast, while CSF1 would appear to be the principal ligand of CSF1R in mammals, the data supported the ancestral pairing of IL34 and CSF1R in early lineages. However, it should also be noted that IL34 has two other receptors, Sydecan (CD138) (*Segaliny et al., 2015*) and protein-tyrosine phosphatase zeta (*Nandi et al., 2013*) and these interactions could also account for the earlier emergence of IL34.

## The emergence of CSF3R/CSF3 and onset of endothermy likely influenced the distribution of neutrophils in Chordates during evolution

The mammalian neutrophil shares many developmental and functional properties with counterparts present in other animal orders such as phagocytosis, oxidative burst, degranulation and cell motility (*Genovese et al., 2013*; *Styrt, 1989*). However; a notable species-specific difference is the distribution of the neutrophil/heterophil within peripheral blood, which suggests that evolutionary pressures might also be involved. To further address this, we reconstructed a set of related timescales plotting the percentage distribution of the chordate myeloid WBCs described in *Figure 1* and the percentage of protein shared sequence similarity of chordate CSFR/CSF described in *Figure 3 versus* time. To simplify the model, the timescales were plotted chronologically, where applicable, and the emergence point of the earliest member within an animal order was used. The timescales were then divided into two distinct periods; the first focussed on the early lineages of jawed/jawless fish, amphibia and early reptilia and the second focussed on all reptilia, aves and mammalia (*Figure 4*).

The synteny studies demonstrated that CSF1R and IL34 were already in existence prior to the appearance of the jawed/jawless fish lineages. However, CSF3R and CSF3 were only present in the Coelacanth and although an IL6-like paralogue was observed in the Whale shark, a cognate receptor was not identified. These results suggested there was a greater level of diversity in the growth factors responsible for WBCs in these lineages and is reflected in the distribution of WBCs in Jawed/jawless fish, where heterophils, neutrophils, monocytes, eosinophils and basophils are all present. While heterophils and monocytes largely dominated, there was a more even distribution of minority populations of neutrophils, basophils and eosinophils (*Figure 4ai*). By the advent of amphibia, distinct neutrophil and heterophil populations had seemingly been lost in favour of a single neutrophil or heterophil population. The amphibian order was the only lineage to demonstrate a largely even distribution of neutrophils, eosinophils, basophils, and monocytes, although there was a slight decline compared to the levels observed in Jawed/Jawless fish (*Figure 4ai*).

At some stage during tetrapod evolution after the emergence of CSF3 and CSF1 haematopoiesis largely transitioned to the tissue-specific compartment of the bone marrow, which likely had implications for WBC distribution in subsequent orders. In early reptilia, represented by the squamata order of lizards, the neutrophil becomes the dominant granulocyte and there is concomitant reduction in basophils and eosinophils. Interestingly, there is an increase in monocyte populations, which coincides with the emergence of the monocyte/macrophage specific growth factor CSF1 (*Figure 4aii*). These changes occur in ectothermic species suggesting that this is more in response to cell-intrinsic factors, rather than external factors such as environment (*Figure 4a*).

The reptilian lineage comprises three orders; squamata, crocodilia and testudines, who emerged over a large timescale. Accordingly, there are many intra-order differences such as the environments in which members live; that is in the water *versus* on land or the presence or absence of limbs or exoskeletons. Interestingly, the neutrophils seemingly peaked in squamata as proportions steadily declined with the lowest levels observed in testudines (*Figure 4bi*). In contrast, eosinophil levels, which remained low across the squamata and crocodilia classes, peaked in testudines. However, across this period in time; CSF1R/CSF1 and CSF3R/CSF3 had become established as blood myeloid cell-specific factors and were unlikely to be driving the changes in WBC distribution (*Figure 4bi*). Notably, as the neutrophil populations declined, both the basophil and eosinophil populations increased, with basophils peaking in the crocodilia lineage before a dramatic decline in testudines. Suggesting, environmental pressure on the immune response of certain orders favoured either the basophil or the eosinophil at the expense of neutrophils and the other granulocyte (*Figure 4bi*). The

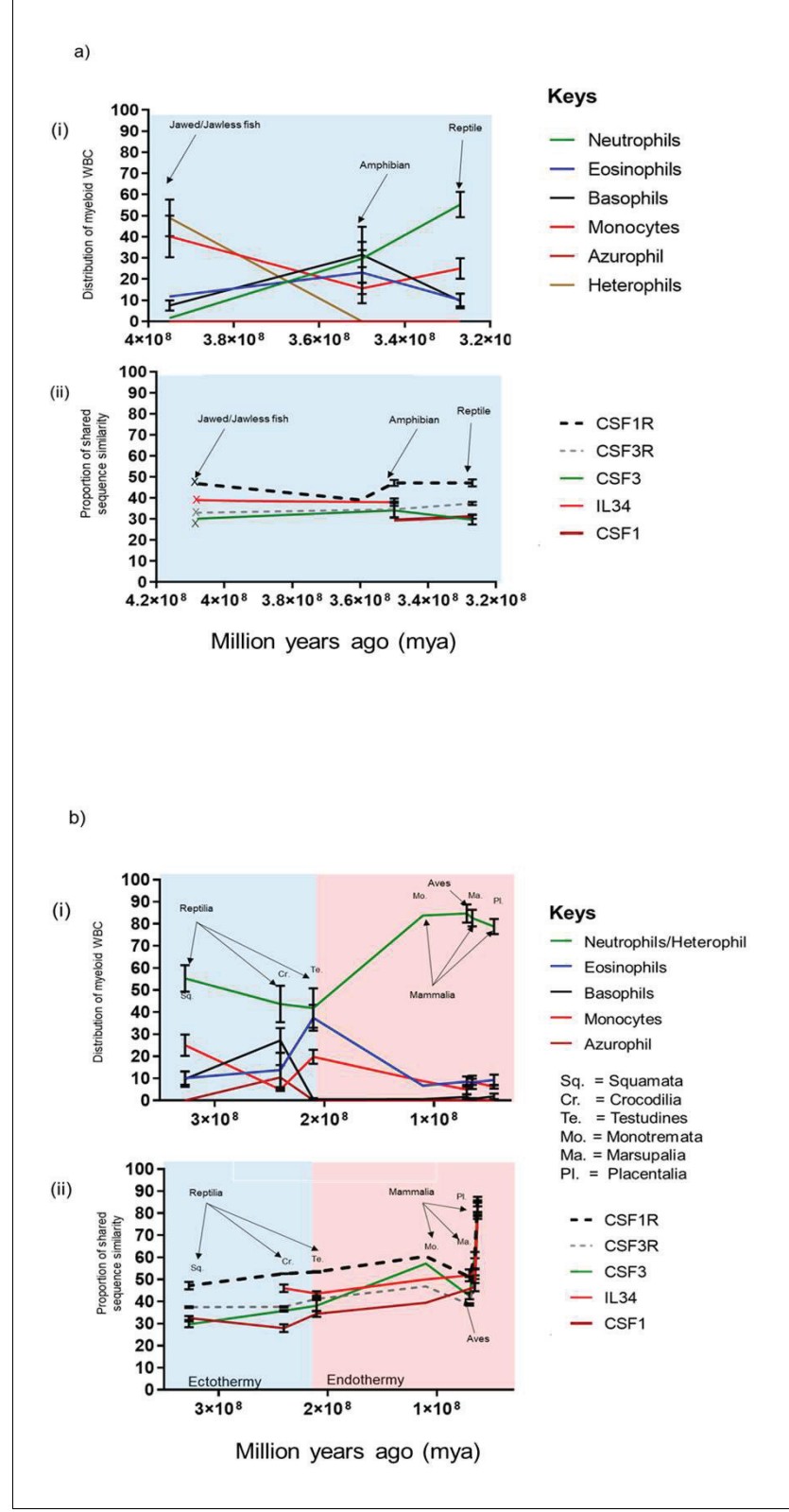

**Figure 4.** Changes in chordate blood granulocyte distribution from Jawed/Jawless lineages through to placental mammals are multi-factorial and likely driven by the emergence of CSF3 and onset of endothermy. (**A**) Graphical representation of % population distribution of myeloid white blood cells *versus* time (i) and % shared sequence similarity of CSF1R and CSF3R protein families *versus* time for Jawed/Jawless fish, Amphibia and the reptilian

*Figure 4 continued on next page*

*Figure 4 continued*

order of Squamata (ii). (B) Graphical representation of % population distribution of myeloid white blood cells *versus* time (i) and % shared sequence similarity of CSF1R and CSF3R protein families *versus* time for the Reptilian orders of squamata, testudines and crocodilia, Aves, and the Mammalian orders of monotremata, marsupalia and Placental (ii). Error bars represent standard deviation.

decline was not only restricted to neutrophil population, as the monocyte population had a similar pattern. Monocytes were at their highest levels in squamata before a rapid decline in crocodilia, however the population expands again in testudines. Notably, as the azurophil emerged in crocodilia, monocytes declined, suggesting that they effectively took over the role of monocytes in crocodilia (*Figure 4bi*).

Interestingly, by the advent of testudines, neutrophil/heterophils were no longer the dominant granulocyte, although granulocytes overall still represented the largest proportion of myeloid WBCs (*Figure 4bi*). However, as the orders of aves, monotremata and marsupalia emerge there is a noticeable change. A limitation of visualising data in this form is the implication of the distinct sequential evolution of different animal orders. However; it's likely there would have been a degree of overlap between the emergence of distinct orders. Therefore, it is noticeable that there was a rapid increase in the both avian heterophil population and monotremata neutrophil population compared to testudines (*Figure 4bi*). The increase in neutrophils was mirrored by an equally rapid decrease in eosinophils in both the aves and mammalian orders (*Figure 4bi*). Neutrophils and eosinophils also share early common progenitors as part of the development pathway in the bone marrow, therefore neutrophil production appeared to be dominating the cell machinery. Interestingly, the switch to the predominance of the neutrophil over the eosinophil, coincides with the emergence of endothermy, strongly suggesting that external factors are behind these changes in myeloid WBC distribution as there are no noticeable changes in the distribution of the CSF1R/CSF1 and CSF3R/CSF3 ligand/receptor pairings. However, the impact is filtering down to the gene/protein level as there are changes in the shared sequence similarity of CSF3, as it increases more rapidly between Testudines and Monotremata, than in the period between Squamata and Testudines. Interestingly, these changes were not reflected in the CSF3R trajectory (*Figure 4bii*). Presumably, the arrival of endothermy resulted in the appearance of novel pathogens for which a neutrophil-mediated response was more appropriate than an eosinophil-mediated one, for example, neutrophils have an array of killing mechanisms such as superior phagocytosis capabilities and preferential production of intracellular respiratory burst that favour intracellular killing of smaller targets (*Borregaard and Cowland, 1997*; *Kovács et al., 2014*; *Hatano et al., 2009*). In contrast, eosinophils favour extracellular killing by deploying granules such major basic protein (MBP) and eosinophil cationic protein (ECP) that are toxic to parasites such as helminths (*Shamri et al., 2011*). These results suggest that in response to the emergence of endothermy and presumed associated novel pathogens, the mammalian host system has selected for the neutrophil as the predominant granulocyte of the blood. The monocyte population had re-emerged as a single population by the appearance of aves and the distribution remained consistent in both the avian and mammalian orders (*Figure 4bi*). Intriguingly, the greatest period of change for myeloid WBC distribution is between testudines and monotremata. However, the equivalent period for protein homology happens much later between the mammalian orders of marsupalia and placentalia (*Figure 4bii*) and is common across all CSFR/CSF pairings. Therefore, as WBC function and distribution had previously been established in early lineages, this suggested that another external factor was responsible for the rapid change, such as the emergence of internalized pregnancy (*Figure 4bii*).

## Discussion

The mammalian neutrophil is a highly specialised cell that acts as a first responder to insults against a host immune system as well as acting in an equally important sentinel role (*Hidalgo et al., 2019*; *Ng et al., 2019*). They are functionally conserved across phylum Chordata and constitute the largest population of myeloid cells in the blood of birds and mammals. In humans, for example, up to one billion neutrophils per kilogram of body weight are produced in the bone marrow each day (*Yvan-Charvet and Ng, 2019*). The immune response has evolved in such a way as to be able to respond

efficiently to a variety of threats, and it is interesting that a resource such as the bone marrow should be expended on the production and maintenance of the relatively short-lived neutrophil at the expense of the eosinophil and basophil. The timescale modelling demonstrates that prior to the advent of tetrapoda lineages, the neutrophil was in a pool of different WBC populations at the disposal of the jawed/jawless fish. However, the appearance of CSF3, altered the distribution and with each emerging animal order, a different granulocyte was favoured, presumably, for cell-specific adaptations in response to environmental challenges. Thus, suggesting that CSFR1R/CSF1 and CSF3R/CSF3 signalling conferred adaptations on the neutrophil that proved evolutionarily advantageous. The work discussed here focusses on the mammalian neutrophil within the context of blood, further work is required to define similar aspects as regards to other tissue resident populations such as macrophages or granulocytes.

The bone marrow is an essential site for granulopoiesis and myelopoiesis. As multipotent haematopoietic stem cells (HSCs) progress through different vascular niches within the bone marrow they sequentially lose their potential to form other lineages in response to environmental cues from bone marrow dwelling macrophages and stromal endothelial cells (*Ng et al., 2019*; *Yvan-Charvet and Ng, 2019*; *Greenbaum and Link, 2011*). The emergence of CSF1 in tetrapod lineages likely led to the appearance of bone and bone marrow. A consequence of this was the gradual re-organisation of haematopoiesis away from existing haematopoietic tissues, such as the eosinophil rich Leydig organ of sharks to tissue-specific compartments in the bone marrow. CSF1, in conjunction with another factor, receptor activator of nuclear factor-κB ligand (RANKL), co-ordinate the reabsorption of old bone through haematopoietically derived osteoclasts to allow the generation of new bone. CSF1 and RANKL (which emerged at a similar point in evolution) orchestrate bone-remodelling through their respective receptors, CSF1R and RANK (*Kim and Kim, 2016*). This suggests that by the time a bone structure had evolved in amphibia, the bone marrow had become the principle site of haematopoiesis, although there are some exceptions within various anurans (*de Abreu Manso et al., 2009*).

In contrast to peripheral blood, CSF3 is expressed on a number of cells in the bone marrow including; neutrophils, monocytes, B cells, myeloid progenitors and HSCs (*Greenbaum and Link, 2011*; *Petit et al., 2002*; *Semerad et al., 2005*; *Lévesque et al., 2003*). Although CSF3 can likely act directly on HSCs through their receptor, it is believed to indirectly mobilise HSCs through a monocytic intermediary that secretes CSF3, which leads to suppression of the CXCR4:CXCL12 axis, alteration of the bone marrow niche, and the subsequent release of HSCs (*Greenbaum and Link, 2011*; *Petit et al., 2002*; *Semerad et al., 2005*; *Lévesque et al., 2003*). In a similar process, CSF3 can suppress B cell lymphopoiesis by again targeting CXCL12 and suppressing other B cell tropic factors or stromal cells that favour the lymphoid niche (*Day et al., 2015*). Thus, the emergence of CSF3R/CSF3 conferred the adaptation, or advantage, of control of the biological machinery i.e. CSF3 provides a mechanism through which haematopoiesis can be shaped and deployed in favour of maximal neutrophil generation. This broadly aligns with the neutrophil starting to dominate peripheral populations in the transition from amphibia to the lizards of early squamata following the appearance of bone.

Mammalian neutrophil production occurs in the haematopoietic cords present within the venous sinuses and the daily output is approximately $1.7 \times 10^9$/kg (*Summers et al., 2010*). As our data demonstrate high levels of neutrophil output are common across warm-blooded chordates and neutrophil/heterophils are the predominant granulocyte in birds and mammals, in what could be considered an example of convergent evolution. One of the fundamental requirements of the host innate immune response is to be able to respond rapidly to perceived threats that can be present at any site in the body, which requires the cellular arm to be constantly present and available. Theoretically this can be achieved in the steady state by having either low volumes of long-lived cells or high volumes of short-lived cells circulating in the periphery. As a consequence, there are potential biological trade-offs when considering each setting, firstly; the longer a cell survives in the periphery, the more effort is required by the host to maintain its survival in terms of providing appropriate cues and growth factors. Secondly, while a short-lived cell does not need as much host input for survival, a high turnover is required in order to ensure it doesn't compete for growth factors with other cell types or cause damage by being retained beyond its usefulness. The latter setting fits with the observed neutrophil life cycle and could be considered an evolutionary adaptation. The CSF3R/CSF3

signalling pathway is essential for generating high neutrophil numbers without adverse effects to the host.

The neutrophil has a short circulatory half-life in the steady state of approximately one day (*Ng et al., 2019*; *Yvan-Charvet and Ng, 2019*), although this does remain controversial as some estimates of the circulatory lifespan are as much as five days (*Pillay et al., 2010*). By contrast, a mature eosinophil has a short circulatory lifespan of approximately 18 hr before relocating to the tissue, where it will survive for a further 2–5 days (*Park and Bochner, 2010*). Similarly, intermediate and non-classical monocytes can survive in the periphery for four and seven days respectively (*Patel et al., 2017*). In human studies, CSF3 has been shown to 'effectively' shorten the lifespan of neutrophil myeloblasts and promyelocytes by decreasing their time spent in cell cycle and accelerating their progress to maturation (*Lord et al., 1989*). Although, it is unclear what the exact effect is on the lifespan of post-mitotic neutrophils, studies have shown that the addition of exogenous CSF3 can delay apoptosis in mature neutrophils (*van Raam et al., 2008*; *Zimbelman et al., 2002*). This suggests that under certain conditions; such as infection, CSF3 can extend the lifespan of a mature neutrophil. Interestingly, classical monocytes also express CSF3R and have a short circulating lifespan of approximately 24 hr (*Patel et al., 2017*) These studies suggest that under steady state conditions and below a certain threshold, CSF3 has an as-yet-undocumented role either directly or indirectly in maintaining a short neutrophil lifespan and thus allowing the efficient turnover required to sustain high levels of granulopoiesis.

CSF3 orchestrates the life cycle of neutrophils in the bone marrow microenvironment by marshalling neutrophil progenitors through different development stages through to maturity, which is indicative of the inductive model of differentiation (*Stanley, 2009*). Accordingly, in response to this, CSF3R is expressed though every life stage, although at differing levels across the neutrophil populations, with the highest levels observed on mature cells where it is expressed at between two and three-fold more than on progenitors (*Demetri and Griffin, 1991*). CSF3R/CSF3 signalling has a dual role in controlling the distribution of neutrophils by both retaining a population of mature neutrophils in the bone marrow as a reservoir and facilitating the egress of other neutrophils into the periphery. Thus, a major evolutionary adaptation that CSF3R/CSF3 has conferred to the neutrophil is the ability to move, both on a population-wide and individual cell level.

There is abundant production of neutrophils daily in the bone marrow; however, there are far fewer neutrophils in circulation in the blood than are produced during granulopoiesis as the total neutrophil population is effectively stored in the bone marrow or marginated in intravascular pools within the spleen and liver (*Ussov et al., 1995*). Neutrophils can transit between sites in response to CSF3 signalling and it is estimated that 49% of cells are present in the circulating pool and the remaining 51% are marginated in discrete vascular pools (*Summers et al., 2010*; *Athens et al., 1961*). In the event of an infection, neutrophils can be mobilised from the marginated pools and bone marrow in response to CSF3-induced production of mobilising signals such as CXCL1 (*Köhler et al., 2011*). The co-ordinated egress of neutrophils from the bone marrow is achieved by CSF3 interacting with the CXCR4/CXCL12 and CXCR2/CXCL2 signalling pathways. CSF3 disrupts the CXCR4-CXCL12 retention axis by reducing CXCL12 release from endothelial stromal cells and reducing CXCR4 expression on neutrophils, thus allowing the movement of mature neutrophils through the venous sinuses to the periphery. CSF3-induced expression of CXCR2 on neutrophils then causes their migration to the vasculature along a CXCL2 chemotactic gradient (*Adrover et al., 2019*; *Eash et al., 2010*; *Eash et al., 2009*).

As neutrophils enter the bloodstream, they need to be able to migrate easily around the circulatory system under high flow conditions. This requires the neutrophil's physical form to have high deformability and flexibility as it encounters the different diameters of the vasculature. The neutrophil nucleus is functionally adapted to this role because of its multi-nucleated structure and the distinct protein composition of the nuclear envelope, features that are widely conserved across mammalian species (*Manley et al., 2018*). CSF3 in co-ordination with C/EBPε and the ETS factors; Pu.1 and GA-binding protein (GABP) are responsible for the transcriptional control of the essential neutrophil nuclear structural proteins; Lamin A, Lamin C and Lamin B receptor (LBR) (*Malu et al., 2016*; *Cohen et al., 2008*). In comparison to other cell nuclear protein compositions, neutrophils have a low proportion of Lamin A and Lamin C, which is believed to make the nucleus more flexible for easier transit (*Manley et al., 2018*; *Malu et al., 2016*; *Cohen et al., 2008*). In contrast, the levels of LBR are increased, which is required for nuclear lobulation and subsequent neutrophil maturation

(*Manley et al., 2018*; *Malu et al., 2016*). However, recent work has also demonstrated that band cells (immature neutrophils) are capable of just as efficient migration with incomplete nuclear segmentation as mature neutrophils (*van Grinsven et al., 2019*). Taken together, these factors support the argument that CSF3 signalling and C/EBPε are intrinsic to the evolution of mammalian neutrophil cell motility.

The evolution of CSF3R/CSF3 has been essential to the development of the neutrophil/heterophil in chordates and its own existence as the principle neutrophil growth factor is evolutionarily advantageous. Some members of the jawed/jawless lineages are unique in that they have populations of both neutrophils and heterophils, whereas later linages favour either neutrophils or heterophils. Our analysis shows that CSF3R/CSF3 emerged before the advent of jawed/jawless fish as both are present in the Coelacanth and absent from the other lineages. Intriguingly, an IL6-like gene was present in the same syntenic location of the whale shark, suggesting the possibility that heterophils and neutrophils were independently controlled by an IL6-like protein and CSF3 in the early lineages. IL6 is an important pleiotropic pro-inflammatory cytokine that plays key roles in infection, inflammation and haematopoiesis (*Tanaka et al., 2014*). Although CSF3R and IIL6R, which are functional paralogues, diverged from each other many millions of years ago, there is still functional redundancy at the cell level in modern-day mammals, as neutrophils are present - though at vastly reduced numbers - in CSF3$^{-/-}$ and CSF3R$^{-/-}$ mice (*Liu et al., 1996*; *Lieschke et al., 1994*). IL-6 can act on neutrophil progenitors and immature neutrophils thus supporting neutrophil development in the bone marrow of CSF3-deficient mice (*Basu et al., 2000*). However, mature neutrophils are refractory to IL-6 signalling as the expression of the IL-6R subunit gp130 is lost during maturation (*Wilkinson et al., 2018*). From amphibia onwards, the protein analysis here suggests that CSF3R and CSF3 co-evolved closely together and in contrast to CSF1R/CSF1/IL34, CSF3 is likely the only ligand of CSF3R.

Using various in-silico approaches, these findings have demonstrated how essential CSF3R and CSF3 (and to a lesser extent CSF1R and CSF1) are to the predominance of mammalian neutrophils in blood. Through the course of evolution, CSF3R/CSF3 signalling has accrued many properties that are responsible for the survival of the neutrophil and its ability to function, such as cell motility and mobility. Interestingly, although CSF3 is present in jawed/jawless fish lineages, it's not until the emergence of tetrapoda that CSF3R and CSF3 begin to dominate haematopoiesis. Our current studies have been limited to chordates in which the data is publicly available and as the datasets continue to be expanded and updated, our hypothesis may need to change to reflect that. However, we would argue that this approach is a valid method for exploring the origins of the neutrophil granulocyte and would further be supported by the physical isolation and characterisation of neutrophils, associated genes and proteins in the different animal orders within Phylum Chordata.

We also envisage exploring the phagocytic granulocyte of two model species. The amphioxus is considered the basal chordate and a macrophage-like population has been identified, although a neutrophil/heterophil has yet to be described within the rudimentary circulatory system (*Han et al., 2010*). The lungfish, a primitive airbreathing fish, is unique among jawed/jawless lineages as it lives in freshwater and can survive on the land for up to one year. Accordingly, the lungfish has many adaptations and is considered to be a one of the closest living relatives to tetrapods making it a good model species for further study (*Takezaki and Nishihara, 2017*). Given how essential motility and mobility are to neutrophil function and development, it would be useful to discern when it emerged in evolution by identifying if equivalent cells and functional gene/protein orthologues are present in either species. These comparative studies would answer fundamental questions about the origin of the neutrophilic phagocyte.

## Materials and methods

### Species selection

Ninety-four animals from all the major classes were selected, thirty-five of which, were used for the calculation of haematological parameters and the remaining fifty-nine for the bioinformatics-based studies. Analyses were performed on non- mammalian lineages including; jawed/jawless fish, amphibia, non-avian reptiles – both non-crocodilian and crocodilian-, aves, and the mammalian lineages; monotremata, marsupialia and placentalia. Urochordates (tunicata), and cephalochordate (Amphoxi) were excluded from analysis as there was insufficient coverage or insufficient annotation

of sequence data in the Pubmed Gene and Ensembl databases. Similarly, the lungfish (Dipnoi) was also excluded from this analysis because of insufficient coverage of sequence data. Finally, teleost lineages were excluded from the analysis owing to having gone through three rounds of genome wide duplication, in contrast to all other chordates who have only undergone two rounds (*Glasauer and Neuhauss, 2014*). A species tree was generated using the NCBI Common Taxonmy browser common tree tool (*Sayers et al., 2009*; *Benson et al., 2009*) and visualised using the Interactive Tree of Life (iTOL) web browser tool (*Letunic and Bork, 2019*).

## Comparative analysis of haematological parameters and CSF/CSFR gene presence and synteny in chordates

Complete blood count (CBC) data for thirty-eight species representing each animal order or class were collated from existing literature to perform a meta-analysis of myeloid WBC in phylum Chordata. Where available, a representative data for each gender within in species class or order were used. Only counts for sub-adult or adults that were calculated to the standardised concentration ($10^9$/L) or equivalent were used and all lymphoid data was excluded (references for CBC data are listed in *Supplementary file 3*). The proportional composition for each myeloid subset; neutrophil, eosinophil, basophil and monocytes as part of the overall myeloid population was also calculated.

Gene sequence for fifty-nine species - representing different animal orders and classes - were retrieved from the NCBI Gene databases (*Pruitt et al., 2007*) in most instances or from annotated entries in Ensembl (v95) (*Yates et al., 2020*) for the following genes; CSF1R, IL34, CSF1, CSF3R, CSF3, C/EBPα, C/EBPβ and C/EBPε (gene identification numbers for species examined are listed in *Supplementary file 4*). A heatmap visualising gene presence or absence was generated using maplotlib/seaborn library for Python package 3.7.7.

Syntenic maps were generated manually for selected genes in the Jawed fish lineage. To generate the maps, syntenic blocks were identified as a section of the human chromosome containing the gene of interest flanked by *x* number of genes. The blocks were then manually compared to similar regions in the chromosomes in Jawed fish to identify orthologous genes. Between three and five genes were chosen per syntenic block to function as anchor points of reference. Anchor points were selected based on their situational proximity, either upstream or downstream, to gene of interest and were present in all species examined. Multiple genes were identified as anchor points to mitigate for the random loss of genes during the process of evolutionary gene rearrangement.

## Generation of percentage shared sequence similarity plots and timescales

As with the gene sequences, protein sequences for the identified species were retrieved from the NCBI Protein databases. Presumed orthologous sequences were screened using the NCBI Basic Local Alignment Search tool (BLAST) [RRID:SCR_004870] to generate a percentage score of sequence similarity (protein accession numbers for species examined are listed in *Supplementary file 5*). The parameters to determine shared sequence similarity were as follows; a sequence was deemed to be homologous where coverage of the protein sequence was equal to or greater than 40% of the total protein sequence examined and the e value was between $0 \times 10^{-20}$ - 0. Percentage scores were averaged per order. Shared sequence similarity was also plotted against animal order or class. The same data was also used to generate two sets of timescales for each order/member for a given receptor/ligand family, where either the calculated mean shared sequence similarity value or haematological parameters were plotted on a timescale based on the emergence of the earliest known ancestor of that order *versus* time (million years ago [mya]).

## Additional information

### Funding

| Funder | Grant reference number | Author |
|---|---|---|
| Medical Research Council | MR/M003159/1 | Kevin J Woollard |
| Kidney Research UK | RP_019_20160303 | Kevin J Woollard |
| Kidney Research UK | RP_002_20170914 | Kevin J Woollard |

| British Heart Foundation | PG/18/41/33813 | Kevin J Woollard |

The authors declare that there was no direct funding for this work. Grants from MRC (MR/M003159/1), Kidney Research UK (RP_019_20160303, RP_002_20170914) and BHF (PG/18/41/33813) support the Woollard lab. KJW is now an employee for AstraZeneca (BioPharmaceuticals R&D, Cambridge, UK). All of this work was performed at Imperial College London. No funding or support was received from AstraZeneca.

## Author contributions

Damilola Pinheiro, Conceptualization, Formal analysis, Investigation, Visualization, Methodology, Writing - original draft; Marie-Anne Mawhin, Conceptualization, Visualization, Writing - review and editing; Maria Prendecki, Conceptualization, Writing - review and editing; Kevin J Woollard, Conceptualization, Supervision, Methodology, Writing - review and editing

## Author ORCIDs

Damilola Pinheiro (ID) https://orcid.org/0000-0003-0294-9423
Maria Prendecki (ID) http://orcid.org/0000-0001-7048-7457
Kevin J Woollard (ID) https://orcid.org/0000-0002-9839-5463

## Decision letter and Author response

Decision letter https://doi.org/10.7554/eLife.60214.sa1
Author response https://doi.org/10.7554/eLife.60214.sa2

# Additional files

## Supplementary files

• Supplementary file 1. Gene and related protein names for the CSF1R/IL34/CSF1 and CSF3R/CSF3 families.

• Supplementary file 2. Knock-out mouse models in *cebpα*, *cebpβ* and *cebpε* deficient mice.

• Supplementary file 3. References for Chordate haematological parameters.

• Supplementary file 4. Gene Identification numbers for species examined.

• Supplementary file 5. Protein accession numbers for species examined.

• Transparent reporting form

## Data availability

All data generated or analysed during this study are included in the manuscript and supporting files.

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
