## [Decision Letter]

**Acceptance summary:**

In this study, Pinheiro and colleagues examine how neutrophils and other myeloid cells evolved. Using a wide range of relational taxonomic data, the authors show how pairings of genes in the granulocyte colony-stimulating factor receptor family (CSF3R/CSF3) and in the macrophage colony-stimulating factor receptor family (CSF1R/IL34/CSF1) evolved and contributed to granulocyte adaptations. This work sheds light into the evolution of mammalian neutrophils.

**Decision letter after peer review:**

Thank you for submitting your article "Analysis of receptor-ligand pairings and distribution of myeloid subpopulations across the animal kingdom reveals neutrophil evolution was facilitated by colony-stimulating factors" for consideration by *eLife*. Your article has been reviewed by three peer reviewers, one of whom is a member of our Board of Reviewing Editors, and the evaluation has been overseen by Patricia Wittkopp as the Senior Editor. The following individuals involved in review of your submission have agreed to reveal their identity: Leo Koenderman (Reviewer #1); Clare Pridans (Reviewer #2).

The reviewers have discussed the reviews with one another and the Reviewing Editor has drafted this decision to help you prepare a revised submission.

Summary:

Pinheiro and colleagues have described a fascinating view on the evolution of neutrophils and other myeloid cells. The authors used a wide range of relational taxonomic data to show how CSF1/CSF1R and CSF3/CSF3R pairings evolved and contributed to granulocyte adaptations. This is a very original and potentially important piece of work that sheds light into the evolution of mammalian neutrophils.

Essential revisions:

1) First of all Figure 4, is missing. This is an important figure as the text (as it is now) is difficult to understand.

2) At several locations in the article the authors imply that G-CSF is inducing differentiation fitting with an inductive model (e.g. Introduction). At the same time the authors rightly mention the presence of mature neutrophils in G-CSF^-/-^ mice (as well as mature eosinophils in IL5R^-/-^ mice) more pointing at a stochastic model. This latter model assumes that expression of CSF-R's is more random, and only committed progenitors expressing these receptors will proliferate rather than differentiate in response to these CSF. Please provide sufficient arguments for the inductive model or change part of the interpretations when a stochastic model is more likely.

3) In the whole article data are provided on numbers in peripheral blood. Only a minority of myeloid cells reside in the blood, the majority is in the tissues. The situation with neutrophils is uncertain. Please discuss.

4) The part on C-EBP transcription factors is difficult to follow. Please help the reader understand why they are so important (based on KO strategies) while there is no clear picture in evolution as the genes are sometimes present, sometimes not. Some species have many some only one. Simply stating redundancy in the system does not really fit the knock-out studies.

5) The part described in the subsection “The emergence of CSF3R/CSF3 and onset of endothermy likely influenced the distribution of neutrophils in Chordates during evolution”/Supplementary Figure 1 is not adding much to the article. It is only human with no evolutionary perspective. Consider removing.

6) Please provide some more insight into the issue of eosinophils versus neutrophils. Now it is implied that the co-evolution with endothermia is relevant. Many articles suggest that eosinophils are more specialized in killing large targets (extracellular killing/e.g. parasites) vs. neutrophils small targets (intracellular killing/e.g., bacteria). Can the authors provide their ideas about the functional difference of the cells in the evolutionary perspective.

7) Discussion. It is stated that neutrophils comprise the largest population of myeloid cells in mammals. This needs supportive evidence, as macrophages are thought to be the largest population at least in the tissues.

8) Discussion. Although the issue of the lamins is well taken formal proof that the segmented nuclear morphology of neutrophils is important for movement and trans-cellular migration is yet to be determined (e.g. van Grinsven et al., 2019 ).

9) Introduction. Young children with SCN often have mutations in the ELANE gene rather than the GSF-R gene. Can the authors discuss how ELANE fits with the model they are presenting?

10) Please provide the definitions of neutrophils and heterophils as they can be present as different cells in the same species.

11) Please provide a supplementary file containing all the references used for Figure 1H (complete blood count data; CBC). This would be a useful source of data for researchers interested in other blood cell types.

12) Regarding the CBC data – the authors should mention in the text if all the samples were obtained from adults. While we appreciate that n values are low for some species, do you obtain the same result if you analyse males and females separately? This may be worth mentioning given that neutrophil numbers have been reported to be higher in women.

13) Please provide a supplementary file containing all the NCBI gene and Ensembl accession numbers for each gene, in each species (Figure 2A).

14) The authors may want to mention that there are other receptors for IL-34 which may explain its expression (in fish, Figure 2A) in the absence of Csf1r.

15) Please provide a supplementary file containing all the NCBI protein accession numbers used for Figure 3A.

16) Please include isotype controls on histogram in Supplementary Figure 1A, C and D.

17) Please include full gating strategy for Supplementary Figure 1A.

18) Why was 72h chosen for the mobility assays (Supplementary Figure 1B)? At this point, monocytes cultured in CSF1 would begin differentiating into macrophages, and this may affect their mobility.

19) Supplementary Figure 1C – please include the antibodies in the Lin cocktail for flow cytometry in the figure legend.

20) Please mention in text and figure legend that human blood was used (there is no mention of it within text).

21) Was a dead cell exclusion dye used for flow cytometry of human blood and neutrophils? And did you look at FSC-A v FSC-H to exclude doublets? If not, how can you exclude the possibility that the Cxcr4 hi neutrophils are not dying or doublets?

22) Since this is a large selection of taxa groups, can specification (of a subset) be divided into more detail?

23) It would be helpful to check the sequence conservation of the receptors across these taxonomic families and see whether there are any minor evolution instances where they mutated. If the receptors have mutated, do they have a particular residue that mutated?

---

## [Author Response]

Essential revisions:1) First of all Figure 4, is missing. This is an important figure as the text (as it is now) is difficult to understand.

We thank the reviewer for bringing this to our attention. This was a problem with transfer from BioXriv and *eLife* (pre-print copy had Figure 4 included). We have now resolved the issue with editorial team and Figure 4 is present in its entirety.

2) At several locations in the article the authors imply that G-CSF is inducing differentiation fitting with an inductive model (e.g. Introduction). At the same time the authors rightly mention the presence of mature neutrophils in G-CSF^-/-^ mice (as well as mature eosinophils in IL5R^-/-^ mice) more pointing at a stochastic model. This latter model assumes that expression of CSF-R's is more random, and only committed progenitors expressing these receptors will proliferate rather than differentiate in response to these CSF. Please provide sufficient arguments for the inductive model or change part of the interpretations when a stochastic model is more likely.

We thank the reviewers for raising these interesting points. As highlighted, functional redundancy is a prominent feature of myelopoiesis, as in the knock-out models identified, IL-6 and CSF2 can both fully or partially function as surrogates for either CSF3, IL5 or both [1, 2] (all references at end of this document). Although this is indicative that redundancy favours the stochastic model, we would argue in homeostatic conditions CSF3 favours an inductive model, as it is exclusively associated with neutrophil development and function. Competitive repopulation assays in CSF3R null mice have shown that CSF3 is responsible for almost all basal granulopoiesis including the regulation, production and maintenance of committed myeloid progenitors [3]. Similarly, in vitro CSF3 treatment of colony forming units -granulocyte, monocyte (CFU-GM), which are capable of producing monocyte or neutrophil populations, resulted in the uniform production of neutrophils, which is supportive of the inductive model [4, 5]. These studies also support a similar model where CSF3 is considered an instructive cytokine as opposed to permissive [6]. The bioavailability of CSF3R could be considered another indicator of the inductive model, as although it is also expressed on haematopoietic stem cells (HSCs), higher levels of expression are restricted to specific neutrophil progenitors increasing through to maturity [7, 8]. Furthermore, in basal conditions, CSF3 is present at higher concentrations in blood serum than CSF2, therefore the CSF3/CSF3R axis “effectively” outcompetes the CSF2/CSF2Ra/CSF2rb axis [8]. Within this context, it could be argued that the elevated presence of CSF3/CSF3R as the default or principle mechanism for neutrophil generation, in mammals at least, strongly supports an inductive model of development. We have now added a new sentence to reflect this hypothesis (Discussion).

3) In the whole article data are provided on numbers in peripheral blood. Only a minority of myeloid cells reside in the blood, the majority is in the tissues. The situation with neutrophils is uncertain. Please discuss.

We thank the reviewers for highlighting this important point. We agree the majority of myeloid cells are expected to reside within tissue. However, the focus of our work was to examine the blood microenvironment and evolution of bona fide blood myeloid cells. Correlating the evolution of CSFs with heterogeneity of myeloid subsets. We provocatively highlight that adaption of granulocyte like behaviour (migration between tight junctions) may correlate with arrival of lungs, blood vessels and acceleration of microbial infection diversity. More work will now be needed to define evolution of other cells outside the vasculature, such as tissue resident macrophages and bone fide subpopulations of granulocytes, such as neutrophils. We have now added text to reflect this in the Discussion.

4) The part on C-EBP transcription factors is difficult to follow. Please help the reader understand why they are so important (based on KO strategies) while there is no clear picture in evolution as the genes are sometimes present, sometimes not. Some species have many some only one. Simply stating redundancy in the system does not really fit the knock-out studies.

We thank the reviewer for their comments and agree the section needs further clarity. Three members of the C/EBP gene family; C/EBPε, C/EBPβ and C/EBPε have been shown to be required for neutrophil development. C/EBPβ, is the earliest expressed member during neutrophil development and is present on HSCs, common myeloid progenitors (CMPs) and granulocyte monocyte progenitors (GMPs) [9]. C/EBPβ is necessary for the transition from the CMP to GMP and when absent, granulopoiesis is impeded and does not progress beyond the CMP stage [9-13]. C/EBPβ is the next member to be expressed and is present on GMPs through to maturation and is required for emergency and/or cytokine induced granulopoiesis. C/EBPβ-deficient mice failed to mobilise granulocytes in response to systemic fungal infection or cytokine stimuli [9, 14-17]. Interestingly, although C/EBPβ is not required for steady-state haematopoiesis, it has been shown in models to functionally compensate for the loss of C/EBPβ [18]. Finally, C/EBPε is required for the terminal differentiation of granulocytes and is the last of the three to be expressed and is only present on neutrophil-committed promyelocytes until maturation [19-24]. We have now expanded this section of work by including a table (new Supplementary file 2) that further details the C/EBP-related knock-out studies to help the reader understand the relevance of these transcription factors to neutrophil development. We agree that system redundancy is not necessarily an appropriate measure and have re-written the section to reflect that the C/EBP gene family is very strongly evolutionarily conserved across Phylum Chordata, indicative of their importance to neutrophil development (subsection “Interrogation of CSF1/CSF1R and CSF3/CSF3R and C/EBP gene family reveals their loss in the early lineages”).

5) The part described in the subsection “The emergence of CSF3R/CSF3 and onset of endothermy likely influenced the distribution of neutrophils in Chordates during evolution”/Supplementary Figure 1 is not adding much to the article. It is only human with no evolutionary perspective. Consider removing.

We thank the reviewers for their comments and agree that as the work currently stands it does not add much to the in silico work presented. We have decided to remove this work, as more lab work is required to complete the preliminary studies discussed and is out of scope of current in silico studies. We are now seeking grant funding to complete this work.

6) Please provide some more insight into the issue of eosinophils versus neutrophils. Now it is implied that the co-evolution with endothermia is relevant. Many articles suggest that eosinophils are more specialized in killing large targets (extracellular killing/e.g. parasites) vs. neutrophils small targets (intracellular killing/e.g., bacteria). Can the authors provide their ideas about the functional difference of the cells in the evolutionary perspective.

We thank the reviewers for their comments. We have considered how eosinophils and neutrophils are specialized towards intra- and extracellular killing respectively Firstly, neutrophils have an arsenal of primary and secondary granules such as; defensins, cathelicidins and lysozymes, which have known antimicrobial properties [25]. In contrast, although eosinophils have a similar set up of primary and secondary granules, eosinophil cationic protein (ECP) and major basic protein (MBP) are both key for extracellular killing, as they are toxic to large targets such as helminth parasites and hemoflagellates as well bacteria [26]. While both eosinophils and neutrophils use respiratory bursts as a killing function, the mechanisms differ. Eosinophils are known to have more O_2_^-^ species than neutrophils, and this is believed to be required for the extracellular targeting of the respiratory burst, a feature unique to eosinophilic granulocytes and implicated in host-pathogen defences against parasites [27-29]. In contrast, neutrophils favour intracellular respiratory bursts for the destruction of internalised pathogens [27-29]. Interestingly, although eosinophils can phagocytose extracellular *S. aureu*s and *E. coli*, they are unable to kill them as efficiently as neutrophils, who are by far the more efficient cell type [30].

Our analysis suggests that as early as the emergence of amphibians, neutrophils and eosinophils had likely acquired their respective functional properties, however, the population proportions are relatively similar. Therefore, it could be posited that the re-organisation of blood granulocyte populations is linked to the environment and likely corresponds with the type and prevalence of pathogens, i.e. pathogen exposure shapes the host myeloid cell pool and selects for the most appropriate cell type. In some ectothermic lineages, where presumably parasites and viral or bacterial pathogens are more prevalent, there is a near equal distribution of eosinophils and neutrophils. However, it could be hypothesised that following the transition from ectothermy to endothermy, there would have been changes to existing pathogens such as loss of pathogenicity or function as they failed to cross over and infect the newly emerging endothermic species, as evidenced by the inability of some viruses and fungi to infect modern day mammals [31-33]. Ultimately, this “thermic switch” would lead to the emergence of novel pathogenic strains that could survive in endothermic hosts (or both). As neutrophils and heterophils dominate the blood of land-dwelling mammals and birds respectively, this again supports an argument that in response to external selection pressures, the host species are redistributing their myeloid blood cell pool and selecting for neutrophil predominance. We have commented further on this within the manuscript (subsection “The emergence of CSF3R/CSF3 and onset of endothermy likely influenced the distribution of neutrophils in Chordates during evolution”).

7) Discussion. It is stated that neutrophils comprise the largest population of myeloid cells in mammals. This needs supportive evidence, as macrophages are thought to be the largest population at least in the tissues.

We thank the reviewer for their comment. We have adjusted our statement in the manuscript to reflect that it applied to the blood microenvironment only (Discussion), where neutrophils are the largest population of myeloid cells. Also see point 3 above.

8) Discussion. Although the issue of the lamins is well taken formal proof that the segmented nuclear morphology of neutrophils is important for movement and trans-cellular migration is yet to be determined (e.g. van Grinsven et al., 2019 ).

We thank the reviewers for bringing this important and highly relevant article to our attention. It is now evident that immature neutrophils (band cells) can efficiently migrate with incomplete segmentation, suggesting at least, that it is not always a requirement for movement and trans-cellular migration in all neutrophil populations. We have amended the text of the manuscript to reflect this (Discussion).

9) Introduction. Young children with SCN often have mutations in the ELANE gene rather than the GSF-R gene. Can the authors discuss how ELANE fits with the model they are presenting?

We thank the reviewers for their interesting comments, and we have further considered the role of ELANE in the model. The ELANE gene encodes for the neutrophil granule serine protease- neutrophil elastase- and is a negative regulator of CSF3R [34-37]. in vitro studies have shown the ELANE can downregulate CSF3R expression on the surface of mature neutrophils and can abrogate proliferative signals provided by CSF3 for expansion of the relevant committed myeloid progenitors [34, 35]. Similarly, it has also been shown that ELANE can antagonise the in vitro activity of CSF3 [34, 35]. Therefore, ELANE is heavily implicated in the regulation of neutrophil development via CSF3R/CSF3 signalling. During SCN the arrest of neutrophil development at the promyelocyte stage and concomitant apoptosis of those precursors results in acute neutropenia. There are several theories as to how aberrant ELANE expression could be implicated in this. Some studies have shown that improper ELANE expression results in endoplasmic reticulum stress (ER stress) which can lead to subcellular mislocalization and subsequent cell death of progenitors [36, 37]. Alternatively, other groups posit that disrupted or truncated CSF3R/CSF3 signalling as mediated by the aberrant expression of mutated ELANE could be a possible mechanism for the subsequent loss of neutrophils [34, 35]. A recent study that used an inducible expression system in mice, showed that a specific ELANE mutation diminished enzymatic activity which resulted in impaired granulocytic differentiation and reduced expression of key neutrophil associated genes such as GFi1, C/EBPδ, C/EBPε and CSF3R [37]. Taken together, these studies suggest there could be multiple mechanisms through which ELANE can dysregulate granulopoiesis and is likely mutation specific. ELANE fits into our model as the disruption caused be improperly functioning ELANE to granulopoiesis is significant as it directly impacts on CSF3R/CSF3 signalling and through this has downstream consequences for the development of neutrophils from committed progenitors. We have added a line to reflect that mutations in the ELANE gene are also involved in SCN pathogenesis (Introduction).

10) Please provide the definitions of neutrophils and heterophils as they can be present as different cells in the same species.

We thank the reviewer for their comments. Heterophils are described as a phagocytic leukocyte that is functionally analogous to the mammalian neutrophil. Heterophils can be distinguished from neutrophils using Romanowsky staining techniques as they do not stain neutral because of the presence of red -orange eosinophilic cytoplasmic granules [38, 39] A definition has been added to manuscript (Introduction).

11) Please provide a supplementary file containing all the references used for Figure 1B (complete blood count data; CBC). This would be a useful source of data for researchers interested in other blood cell types.

We thank the reviewer for their comments. A supplementary file (Supplementary file 3) has been added that contains all the references included in the meta-analysis.

12) Regarding the CBC data – the authors should mention in the text if all the samples were obtained from adults. While we appreciate that n values are low for some species, do you obtain the same result if you analyse males and females separately? This may be worth mentioning given that neutrophil numbers have been reported to be higher in women.

We thank the reviewer for their comments and have clarified within the Materials and methods section, that datasets used for the metanalysis used sub-adult and adult data and were pooled with a minimum of male and female set where the data was available. It was not possible to take this approach for every species considered and within the context of this current study it is not possible to comment definitively on if there are gender-specific differences observed.

13) Please provide a supplementary file containing all the NCBI gene and Ensembl accession numbers for each gene, in each species (Figure 2A).

We thank the reviewer for their comments. A supplementary file (Supplementary file 4) has been added that contains all the ID numbers for each gene, in each species examined.

14) The authors may want to mention that there are other receptors for IL-34 which may explain its expression (in fish, Figure 2A) in the absence of Csf1r.

We thank for the reviewer for bringing these receptors to our attention. We have now included a line referencing the IL-34 receptors, Protein tyrosine phosphatase zeta and CD138 (Sydecan), but have cited this in the general discussion regarding the emergence and co-evolution of CSF1R and IL34 (subsection “Analysis of Chordate orthologous protein homology further supports the ancestral pairing of CSF1R/IL34 and CSF3R/CSF3 in early lineages”).

15) Please provide a supplementary file containing all the NCBI protein accession numbers used for Figure 3A.

We thank the reviewer for their comments. A supplementary file (Supplementary file 5) has been added that contains all the ID numbers for each protein, in each species examined.

16) Please include isotype controls on histogram in Supplementary Figure 1A, C and D.

We thank for the reviewer for their comments, this section of work has now been removed.

17) Please include full gating strategy for Supplementary Figure 1A.

We thank for the reviewer for their comments, this section of work has now been removed.

18) Why was 72h chosen for the mobility assays (Supplementary Figure 1B)? At this point, monocytes cultured in CSF1 would begin differentiating into macrophages, and this may affect their mobility.

We thank for the reviewer for their comments, this section of work has now been removed.

19) Supplementary Figure 1C – please include the antibodies in the Lin cocktail for flow cytometry in the figure legend.

We thank for the reviewer for their comments, this section of work has now been removed.

20) Please mention in text and figure legend that human blood was used (there is no mention of it within text).

We thank for the reviewer for their comments, this section of work has now been removed.

21) Was a dead cell exclusion dye used for flow cytometry of human blood and neutrophils? And did you look at FSC-A v FSC-H to exclude doublets? If not, how can you exclude the possibility that the Cxcr4 hi neutrophils are not dying or doublets?

We thank for the reviewer for their comments, this section of work has now been removed.

22) Since this is a large selection of taxa groups, can specification (of a subset) be divided into more detail?

We thank the reviewers for their comments. Where applicable larger taxonomic groups have been subdivided into smaller groups such as reptilia being divided in squamata, crocodilia and testudines. Similarly, mammalia has also been subdivided into the respective groups of monotremata, marsupalia and placentalia. For the purpose of these studies, the cell population of interest does not further subset as determined by CSF3R.

23) It would be helpful to check the sequence conservation of the receptors across these taxonomic families and see whether there are any minor evolution instances where they mutated. If the receptors have mutated, do they have a particular residue that mutated?

We thank the reviewers for this good suggestion. However, we feel that this is beyond the scope of the original focus of the study. The work here addresses changes on a species or sub-species wide scale and how this might have contributed overall to the functional properties of mammalian neutrophils. Although the suggested idea is very interesting, we feel that focussing on changes at the residue level would not really fit in with the wider macro level view and might confuse the overall message.

**References**

1) Walker, F., et al., IL6/sIL6R complex contributes to emergency granulopoietic responses in G-CSF– and GM-CSF–deficient mice. Blood, 2008. 111(8): p. 3978-3985.2)Kopf, M., et al., IL-5-deficient mice have a developmental defect in CD5^+^ B-1 cells and lack eosinophilia but have normal antibody and cytotoxic T cell responses. Immunity, 1996. 4(1): p. 15-24.3) Richards, M.K., et al., Pivotal role of granulocyte colony-stimulating factor in the development of progenitors in the common myeloid pathway. Blood, 2003. 102(10): p. 3562-3568.4) Sonoda, Y., et al., Analysis in serum-free culture of the targets of recombinant human hemopoietic growth factors: interleukin 3 and granulocyte/macrophage-colony-stimulating factor are specific for early developmental stages. Proceedings of the National Academy of Sciences, 1988. 85(12): p. 4360.5) Avalos, B.R., Molecular analysis of the granulocyte colony-stimulating factor receptor. Blood, 1996. 88(3): p. 761-777.6) Stanley, E.R., Lineage Commitment: Cytokines Instruct, At Last! Cell Stem Cell, 2009. 5(3): p. 234-236.7) Demetri, G.D. and J.D. Griffin, Granulocyte colony-stimulating factor and its receptor. Blood, 1991. 78(11): p. 2791-808.8) Lee, K.Y., et al., Varying expression levels of colony stimulating factor receptors in disease states and different leukocytes. Experimental and Molecular Medicine, 2000. 32(4): p. 210-215.9) Scott, L.M., et al., A novel temporal expression pattern of three C/EBP family members in differentiating myelomonocytic cells. Blood, 1992. 80(7): p. 1725-35.10) Smith, L.T., et al., PU.1 (Spi-1) and C/EBP α regulate the granulocyte colony-stimulating factor receptor promoter in myeloid cells. Blood, 1996. 88(4): p. 1234-47.11) Ma, O., et al., Granulopoiesis requires increased C/EBPα compared to monopoiesis, correlated with elevated Cebpa in immature G-CSF receptor versus M-CSF receptor expressing cells. PLoS One, 2014. 9(4): p. e95784.12) Zhang, P., et al., Enhancement of hematopoietic stem cell repopulating capacity and self-renewal in the absence of the transcription factor C/EBP α. Immunity, 2004. 21(6): p. 853-63.13) Ford, A.M., et al., Regulation of the myeloperoxidase enhancer binding proteins Pu1, C-EBP α, -β, and -δ during granulocyte-lineage specification. Proc Natl Acad Sci U S A, 1996. 93(20): p. 10838-43.14) Zhang, P., et al., Induction of granulocytic differentiation by 2 pathways. Blood, 2002. 99(12): p. 4406-12.15) Hirai, H., et al., C/EBPbeta is required for 'emergency' granulopoiesis. Nat Immunol, 2006. 7(7): p. 732-9.16) Screpanti, I., et al., Lymphoproliferative disorder and imbalanced T-helper response in C/EBP β-deficient mice. EMBO J, 1995. 14(9): p. 1932-41.17) Tanaka, T., et al., Targeted disruption of the NF-IL6 gene discloses its essential role in bacteria killing and tumor cytotoxicity by macrophages. Cell, 1995. 80(2): p. 353-61.18) Jones, L.C., et al., Expression of C/EBPbeta from the C/ebpalpha gene locus is sufficient for normal hematopoiesis in vivo. Blood, 2002. 99(6): p. 2032-6.19) Yamanaka, R., et al., Impaired granulopoiesis, myelodysplasia, and early lethality in CCAAT/enhancer binding protein epsilon-deficient mice. Proc Natl Acad Sci U S A, 1997. 94(24): p. 13187-92.20) Morosetti, R., et al., A novel, myeloid transcription factor, C/EBP epsilon, is upregulated during granulocytic, but not monocytic, differentiation. Blood, 1997. 90(7): p. 2591-600.21) Verbeek, W., et al., C/EBPepsilon -/- mice: increased rate of myeloid proliferation and apoptosis. Leukemia, 2001. 15(1): p. 103-11.22) Chih, D.Y., et al., Modulation of mRNA expression of a novel human myeloid-selective CCAAT/enhancer binding protein gene (C/EBP epsilon). Blood, 1997. 90(8): p. 2987-94.23) Lekstrom-Himes, J. and K.G. Xanthopoulos, CCAAT/enhancer binding protein epsilon is critical for effective neutrophil-mediated response to inflammatory challenge. Blood, 1999. 93(9): p. 3096-105.24) Gombart, A.F., et al., Aberrant expression of neutrophil and macrophage-related genes in a murine model for human neutrophil-specific granule deficiency. J Leukoc Biol, 2005. 78(5): p. 1153-65.25) Borregaard, N. and J.B. Cowland, Granules of the Human Neutrophilic Polymorphonuclear Leukocyte. Blood, 1997. 89(10): p. 3503-3521.26) Shamri, R., J.J. Xenakis, and L.A. Spencer, Eosinophils in innate immunity: an evolving story. Cell and tissue research, 2011. 343(1): p. 57-83.27) Bolscher, B.G.J.M., et al., NADPH:O2 oxidoreductase of human eosinophils in the cell-free system. FEBS Letters, 1990. 268(1): p. 269-273.28) Lacy, P., et al., Divergence of Mechanisms Regulating Respiratory Burst in Blood and Sputum Eosinophils and Neutrophils from Atopic Subjects. The Journal of Immunology, 2003. 170(5): p. 2670.29) Kovács, I., et al., Comparison of proton channel, phagocyte oxidase, and respiratory burst levels between human eosinophil and neutrophil granulocytes. Free Radical Research, 2014. 48(10): p. 1190-1199.30) Hatano, Y., et al., Phagocytosis of heat-killed *Staphylococcus aureus* by eosinophils: comparison with neutrophils. APMIS, 2009. 117(2): p. 115-23.31) Robert, V.A. and A. Casadevall, Vertebrate Endothermy Restricts Most Fungi as Potential Pathogens. The Journal of Infectious Diseases, 2009. 200(10): p. 1623-1626.32) Chinchar, V.G., Ranaviruses (family Iridoviridae): emerging cold-blooded killers. Archives of Virology, 2002. 147(3): p. 447-470.33) Wirth, W., et al., Ranaviruses and reptiles. PeerJ, 2018. 6: p. e6083-e6083.34) Hunter, M.G., et al., Proteolytic cleavage of granulocyte colony-stimulating factor and its receptor by neutrophil elastase induces growth inhibition and decreased cell surface expression of the granulocyte colony-stimulating factor receptor. Am J Hematol, 2003. 74(3): p. 149-55.35) Piper, M.G., et al., Neutrophil elastase downmodulates native G-CSFR expression and granulocyte-macrophage colony formation. J Inflamm (Lond), 2010. 7(1): p. 5.36) Nayak, R.C., et al., Pathogenesis of ELANE-mutant severe neutropenia revealed by induced pluripotent stem cells. J Clin Invest, 2015. 125(8): p. 3103-16.37) Garg, B., et al., Inducible expression of a disease-associated. J Biol Chem, 2020. 295(21): p. 7492-7500.38) Montali, R.J., Comparative pathology of inflammation in the higher vertebrates (reptiles, birds and mammals). Journal of Comparative Pathology, 1988. 99(1): p. 1-26.39) Horobin, R.W., How Romanowsky stains work and why they remain valuable — including a proposed universal Romanowsky staining mechanism and a rational troubleshooting scheme. Biotechnic and Histochemistry, 2011. 86(1): p. 36-51.